# Light-induced fine-tuning of optical cavities for organic optoelectronic devices

Shen Xing [1] ✉, Eva Bittrich [2], Vasiliki Prifti[1], Stephanie Buchholtz[1], Yuan Liu[3], Louis Conrad Winkler [1], Maximilian F. X. Dorfner[4], Mikhail Malanin[2], Mingchao Wang [5], Guoqin Liu[6], Dinara Samigullina[1], Anna-Lena Hofmann[1], Jakob Wolansky [1], Jörn Vahland [1], Tianyi Zhang [1], Rongjuan Huang[1], Samuel Dominic Seddon[7], Dieter Fischer[2], Sebastian Reineke[1], Frank Ortmann [4], Xinliang Feng [6], Hans Kleemann [1] ✉, Johannes Benduhn [1] ✉ & Karl Leo [1] ✉

Precise structural control is essential for high-performance semiconductors. In organic electronics, traditional methods for tuning the dimensions of device structures often rely on cumbersome, limited-resolution processes such as shadow mask patterning, printing, or viscosity tuning. Here, we report ultra-violet (UV) irradiation in ambient conditions as a transformative approach for tuning structural parameters of organic small molecule hole transport layers (HTLs) in vertical and lateral directions. The method preserves HTL conductivity while facilitating uniform thickness reduction through synergistic photo-induced oligomerization and photo-oxidative layer shrinking. Controlled thinning applies to various organic materials. In cavity architectures, UV-treated organic photodetectors show narrowband detection from 900 to 1200 nm with a full width at half maximum down to 25 nm, and UV-treated organic light-emitting diodes exhibit 75 nm peak tunability. Moreover, this strategy permits micrometer-scale lateral patterning of HTLs. Our work opens new opportunities for precise and practical engineering for organic electronic devices.

Photonic processing techniques, such as photolithography, laser cutting, and ultraviolet (UV) curing, have significantly advanced optoelectronic devices by offering cost-effective, non-contact, and selective methods for enhancing device functionality through controlled manipulation of light-matter interactions[1–9]. A notable example is the use of excimer UV lasers for light-induced rapid annealing, a technique widely employed to crystallize amorphous silicon into crystalline layers. This process is pivotal in the production of low-temperature polysilicon (LTPS) transistors, which are integral to millions of active matrix displays. Building on its success in silicon (Si)-based devices, this approach has inspired the development of perovskite-based devices, aiming to improve film quality by surface passivation[10,11].

In contrast, photonic processing techniques in organic semiconductors remain scarce, particularly under short-wavelength

[1]Dresden Integrated Center for Applied Physics and Photonic Materials (IAPP) and Institute of Applied Physics, Technische Universität Dresden, Nöthnitzer Straße 61, Dresden, Germany. [2]Leibniz-Institut für Polymerforschung Dresden e.V., Division of Macromolecular Chemistry, Hohe Straße 6, Dresden, Germany. [3]Key Laboratory of the Ministry of Education for Optoelectronic Measurement Technology and Instrument, Beijing Information Science & Technology University, No. 12 Xiaoying East Road, Beijing, China. [4]Department of Chemistry, TUM School of Natural Sciences and Atomistic Modeling Center, Munich Data Science Institute, Technische Universität München, 85748 Garching b, München, Germany. [5]State Key Laboratory of Advanced Waterproof Materials, School of Advanced Materials, Peking University, Shenzhen Graduate School, Shenzhen, China. [6]Faculty of Chemistry and Food Chemistry, Technische Universität Dresden, Bergstraße 66, Dresden, Germany. [7]Institute of Applied Physics, Technische Universität Dresden, Nöthnitzer Straße 61, Dresden, Germany. ✉e-mail: shen.xing@tu-dresden.de; hans.kleemann1@tu-dresden.de; johannes.benduhn@tu-dresden.de; karl.leo@tu-dresden.de

irradiation such as UV. The primary reason arises from the relatively low bond dissociation energies of organic molecules: bonds such as C−C ( ~ 3.6 eV) and aromatic C−H ( ~ 4.1 eV) are susceptible to cleavage when exposed to high-energy UV photons (~4.4 eV)[12]. These reactions can alter the chemical structure and lead to changes in optical and electronic properties[13,14]. As a result, UV exposure is often associated with irreversible degradation, including trap formation, morphological changes, and significantly reduced charge transport[15–20].

Despite these challenges, the reactivity of organic materials to UV light may also present opportunities if it can be precisely manipulated for structure tuning. This is particularly valuable in the fabrication of advanced organic electronic devices, including top-emitting organic light-emitting diodes (OLEDs) and organic photodetectors (OPDs) within cavity structures, crucial for applications such as smartphone displays and biometric sensing functions[21,22]. Consistent cavity thickness is vital for optimal display performance and color accuracy in OLEDs, as well as precise detection wavelengths in OPDs. However, traditional methods such as thermal evaporation with shadow or photolithography masks are very limited for precise post-fabrication of organic materials. These techniques typically require additional test structures for precise absolute vertical thickness control, reducing the processing time of the production tool. Secondly, a lateral thickness adjustment to correct for inhomogeneities, e.g., a gradient, is basically not feasible since it would require many additional processing steps. Therefore, there is a strong need for a scalable and precise method that can locally adjust film thickness after deposition while maintaining the material's electronic functionality.

In this work, we report a UV irradiation method in ambient air for treating nine different organic materials. With increased UV irradiation time, a controlled and uniform layer thinning in sub-nanometer-scale is realized. Simultaneously, the conductivity of the investigated hole transport layers (HTLs) remains robust, irrespective of the doping concentration. The synergistic effects are endowed by the photo-induced oligomerization and photo-oxidative layer shrinking, as demonstrated by spectroscopic ellipsometry and chemically characterized by matrix-assisted laser desorption/ionization time-of-flight (MALDI-TOF) mass spectroscopy, X-ray photoelectron spectroscopy (XPS), Raman, as well as Fourier transform infrared spectroscopy (FTIR). This method leverages UV irradiation to induce beneficial structural modifications in HTLs without compromising their electronic functionality. Unlike conventional techniques such as thermal evaporation or spin-coating, it enables controlled, localized post-deposition adjustment with nanometer vertical precision. This capability not only mitigates sensitivity to initial film thickness variations but also facilitates the simultaneous deposition of optical tooling reference and functional devices in the same vacuum condition. UV post-treatment allows precise thickness correction without restarting the fabrication process of functional devices. This approach ensures consistent layer quality and aligns well with established industrial workflows in organic electronics. To demonstrate the practical applicability of this approach, we successfully fabricate OPDs and OLEDs with cavity architectures, which are widely used in near-infrared (NIR) sensors and smartphone displays. The tunability of the response (OPD) / emission peak (OLED), well-maintained device performance (OPD and OLED) and lifetime (OLED) showcase the versatility of this approach. Furthermore, by employing UV irradiation, we achieve structured HTLs with micrometer-scale resolution, presenting great opportunities for precise control, miniaturization, and integration of devices within the burgeoning organic electronics market.

## Results
### Thickness reduction effect and stable electrical properties of UV irradiated organic HTLs
To demonstrate the processing technique, we prepare a series of intrinsic N, N′-((diphenyl-N, N′-bis)9,9,-dimethyl-fluoren-2-yl)-

benzidine (BF-DPB) films, a commonly used HTL depicted in Fig. 1a, along with its p-doped variant, BF-DPB:NDP9 (10 wt.%). These layers are deposited on silicon (Si) / quartz substrates through physical vapor deposition, followed by a time-controlled UV irradiation in the ambient atmosphere. This process uses an amalgam lamp with a UVC irradiated spectrum provided in Supplementary Fig. 1. A schematic treatment procedure and the employed device architectures are illustrated in Fig. 1b. To characterize the optical properties of the irradiated films (absorption stack configuration in Fig. 1b), spectroscopic ellipsometry measurements are performed. For BF-DPB, the in-plane refraction index ($n$) and the extinction coefficient ($k$) exhibit a gradual decrease over increasing irradiation time (Supplementary Fig. 2). Upon introducing the NDP9 dopant, the in-plane optical constants of the film tend to be more stable and reduce its susceptibility to UV exposure (Supplementary Fig. 2). The diminished $k$ value during treatment leads to a drop in the absorption peak of BF-DPB, albeit with a more pronounced reduction ratio (Supplementary Fig. 3). In particular, for UV-treated BF-DPB:NDP9, the discernible disparity in the change of $k$ and absorption intensity suggests additional alterations in film properties due to the UV irradiation (Supplementary Fig. 3). By evaluating the layer thickness from ellipsometry data for an extended set of UV irradiation times up to 12 h, a gradually thinning of the film under UV treatment can be confirmed, characterized by an approximately linear reduction that ultimately leads to complete disappearance (Fig. 1c). Detailed information on the film thickness under UV treatment is available in Supplementary Table 1. The intrinsic BF-DPB layer shows a thickness reduction rate of $24.8 \pm 0.9$ nm h$^{-1}$, which is marginally faster than the doped film BF-DPB: NDP9 ($21.6 \pm 0.6$ nm h$^{-1}$), suggesting a small stabilizing impact of the dopant on the thickness thinning effect. This is likely attributed to improved molecular packing, intermolecular interactions, and reduced molecular mobility induced by NDP9 doping, lowering the susceptibility of BF-DPB to UV. Importantly, if the reduction would be solely due to an etching effect, changes would be observed only in thickness, not in the material's optical constants ($n$ and $k$). However, variations in $n$ and $k$ indicate that the thinning also involves changes at the molecular level or in the nanostructure of the material.

To validate the general applicability of this phenomenon, various HTLs (Fig. 1a), such as N, N, N′, N′-tetrakis(4-methoxyphenyl)-benzidine (MeO-TPD), N, N′-di(naphthalen-1-yl)-N, N′-diphenyl-benzidine (NPB), 2,2′,7,7′-tetrakis-(N, N-diphenylamino)−9,9′-spirobifluoren (Spiro-TAD), 4,4′,4″-tris(carbazol-9-yl)-triphenylamine (TCTA), spiro-tetra(p-methyl-phenyl)-benzidine (Spiro-TTB), 9,9-bis{4-[di-(p-biphenyl)aminophenyl]} fluorene (BPAPF), and electron transport materials like 1,3,5-tris(N-phenylbenzimidazol-2-yl)benzene (TPBi) and buckminster fullerene (C$_{60}$) are investigated in metal-organic-metal cavity structures under UV irradiation. An optical cavity structure of Cr (3 nm) / Au (60 nm) / organic materials (150 nm) and UV-treated for a specific time / Ag (25 nm) allows for easy characterization of layer thickness variations through the corresponding resonance peak shift (cavity structure in Fig. 1b). The increasing deviation from the original peak (without UV treatment) under UV irradiation intuitively signifies the thinning effect (Fig. 1d), underscoring the versatility of this method. Detailed absorption profiles of the investigated optical cavities are depicted in Supplementary Fig. 4.

The pivotal electrical parameter of transport layers is their conductivity ($\sigma$). The intrinsic HTLs, composed of organic small molecules, typically exhibit limited conductivity due to their relatively low charge carrier mobility and low intrinsic charge carrier density[23]. However, by means of efficient molecular doping, $\sigma$ can be enhanced by several orders of magnitude[24], which is of paramount importance for the organic semiconductor technology. As shown in Fig. 1e, and based on the conductivity configuration displayed in Fig. 1b, UV treatment does not severely compromise the electrical properties of HTLs, which is contrary to initial expectations. Notably, $\sigma$ of BF-DPB: NDP9,

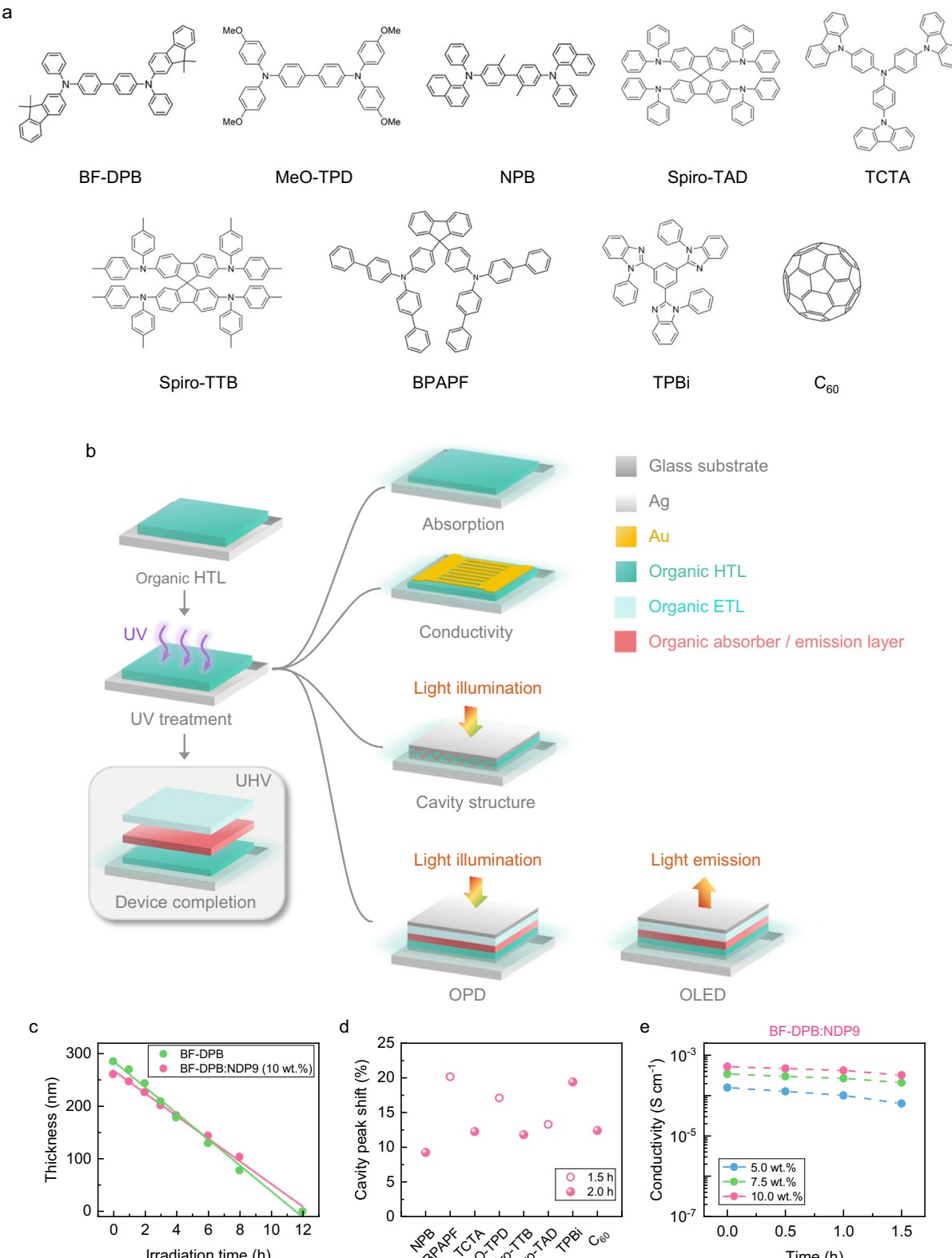

**Fig. 1 | Thickness reduction and constant electrical properties of UV irradiated organic HTLs. a** Chemical structures of the investigated organic materials. **b** Schematic of UV irradiated organic HTL for various measurements, including absorption, conductivity, and thickness reduction under cavity structures, and processing procedures for UV-treated OPDs, UV-treated OLEDs. **c** Thickness of BF-DPB (~ 300 nm) and BF-DPB: NDP9 (10 wt.%, ~300 nm) films as a function of UV irradiation time. **d** Relative shift in absorption peaks for investigated organic materials, including alpha-NPD, TCTA, Spiro-TTB, TPBi, and C$_{60}$ after 2 h of UV exposure (solid circle), and BPAPF, MeO-TPD, and Spiro-TAD after 1.5 h of UV exposure (hollow circle), relative to their original absorption in a Cr (3 nm) / Au (60 nm) / various organic material (150 nm) / Ag (25 nm) cavity structure. **e** Conductivity of BF-DPB: NDP9 films (100 nm) with different doping ratios as a function of UV irradiation time. The conductivity is calculated based on the reduced thickness.

irrespective of doping concentration, exhibits only a mild decrease under UV irradiation. This observation alleviates concerns regarding the vulnerability of HTLs to UV light. For instance, a high $\sigma$ around $4 \times 10^{-4}$ S cm$^{-1}$ can be well maintained after an UV irradiation time up to 1.5 h, demonstrating that the high doping efficiency is not compromised under intense UV conditions. Similarly, a stable $\sigma$ is observed in various NDP9-doped organic films, including MeO-TPD, Spiro-TAD, NPB, and Spiro-TTB under extended UV exposure up to 3.0 h (Supplementary Fig. 5). These results further support the general applicability and robustness of the UV thinning process across different HTL materials. The optical constants are shown in Supplementary Fig. 6, exhibiting minimal variation with irradiation time, in good agreement with findings observed in BF-DPB: NDP9 films. Another dopant, CDX-B, is tested[25], confirming that the stabilization effect is not limited to NDP9 but can be generalized to other dopants with suitable electronic structures (Supplementary Fig. 7). Since these treatments are conducted in air, the results highlight not only the electrical stability but also the air stability of the doped HTLs. This dual stability is essential for their further integration into electronic devices. Additionally, uniform thickness reduction (Supplementary Fig. 8), accompanied by a stable $\sigma$, is obtained on films of 2.5 cm × 2.5 cm. To further evaluate scalability, UV treatment is performed on a 15 cm × 15 cm BF-DPB: NDP9 film. Consistent thickness reduction and a slight decrease in root mean square (RMS) roughness from 0.4 nm to 0.3 nm are observed across the top, center, and bottom positions, as determined by atomic force microscopy (AFM) (Supplementary Fig. 9), confirming uniform treatment over a large area. In addition, the UV irradiation method enables selective treatment of thick regions, eliminating local inhomogeneities and improving overall film uniformity. These results highlight the potential of the UV technique for application in large-scale devices. The reduction rate can be further accelerated in future industrial workflows by employing larger or higher-powered UV sources and multiple exposure stations. Additional mobility measurements of intrinsic BF-DPB and C$_{60}$ under UV/air exposure are presented in Supplementary Fig. 10, confirming moderate sensitivity and supporting the enhanced stability observed in doped systems.

## Prerequisites for initiating thickness thinning effect

To elucidate the mechanism underlying the UV-induced variations in film thickness, a comprehensive experimental study is conducted, focusing primarily on the impact of the incident UV wavelength. Utilizing a BF-DPB:NDP9 (10 wt.%, 150 nm) layer, optical bandpass (BP) filters with center wavelengths of 436 nm, 405 nm, 365 nm, 313 nm, and 254 nm are employed to select the specific UV wavelength, respectively. Moreover, to assess the influence of the 185 nm UVC radiation generated by the amalgam lamp, a longpass (LP) filter with a cutting edge at 235 nm is employed. The transmission characteristics of the filters used in these experiments are detailed in Supplementary Fig. 11. To ensure a well-founded comparison, the number of incident photons is carefully controlled. An optical cavity, identical to the one described above, is constructed to facilitate a detailed examination of the wavelength-dependent effects on film thickness. As depicted in Fig. 2a, the employment of the 254 nm BP filter results in a minimal peak shift, whereas a pronounced 70 nm shift is observed in the films subjected to the LP filter at 235 nm, suggesting a significant enhancement of the UV-induced effect by applying UV-C light only. Remarkably, the maximum blue shift (150 nm) in the absorption peak of the films exposed to unfiltered UV radiation is more than double that observed in the films treated with the LP filter. This observation unravels the critical role of 185 nm deep UV radiation in the thinning process of the films, thereby highlighting the wavelength-dependent nature of the UV-induced thinning effect.

We next investigate the influence of the atmosphere on the UV treatment. All samples embedded in optical cavities with diverse organic materials are therefore irradiated under UV in a nitrogen (N$_2$)

atmosphere. Intriguingly, the absorption peak of the treated sample closely aligns with that of the untreated counterparts, as illustrated in Fig. 2b. This marked deviation from the results obtained in air demonstrates that UV radiation alone does not induce the thinning effect. Instead, it addresses the critical role of oxygen as a catalyst in this process. Detailed absorption spectra of test cavities on studied organic materials in N$_2$ atmosphere are provided in Supplementary Fig. 12. To more directly elucidate the impact of oxygen and ozone, a 100 nm BF-DPB:NDP9 layer is prepared and subjected to ozone, albeit in the absence of UV irradiation. Notably, the film thickness remained unchanged even after 6 h (Supplementary Fig. 13), compellingly supporting the hypothesis of a synergistic interplay between UV irradiation and oxygen presence, which is essential for the initiation of the film thinning effect.

## Mechanism investigation for thickness reduction effect

Since the temperature reached during the UV treatment is far below the glass transition temperature of the material, the potential influence of thermal effect is initially eliminated. The investigation is primarily centered on understanding the effects of UV irradiation on the chemical structure of the organic materials; hence, multiple spectroscopic techniques, including HPLC-MS, XPS, Raman, and FTIR, are employed. MeO-TPD is an ideal candidate due to its similar chemical and electrical properties with BF-DPB, as well as comparable rates of thickness reduction (Supplementary Fig. 14). Furthermore, the symmetric structure and oxygen atoms of MeO-TPD (Fig. 1a) enable a detailed exploration of its structural evolution. We initially prepared a 150 nm MeO-TPD film for HPLC-MS analysis (Supplementary Fig. 15). Following 3 h of UV exposure, a portion of MeO-TPD molecules with a mass-to-charge ratio (m/z) of 609 remain, supporting effective charge transport. Concurrently, a significant decrease in peak intensity and the appearance of peaks at higher masses suggest structural changes, such as oligomerization. To further probe these structural modifications, we conducted XPS measurements on a 40 nm MeO-TPD film. Following 0.5 h of UV irradiation, the N1s spectrum of treated MeO-TPD shows a subtle peak broadening attributed to alterations in the bonding environment proximate to the nitrogen atoms; however, the overall profile keeps constant (Fig. 2c, bottom), indicating the preservation of C−N bonds and[26,27], by extension, the structural integrity of the aromatic rings, thereby maintaining electric conductivity. The post-treatment O1s spectrum can be deconvoluted to three components peaking at 531.8 eV, 533.0 eV, and 534.1 eV, representing hydroxyl (OH) groups, C−O−C (aromatic rings) bonds and likely ester (O−C = O) or (C = O)−O−(C = O) groups (Fig. 2d, bottom)[28,29]. This suggests that UV exposure may facilitate the breakdown of C-H bonds in side chains, reacting with atmospheric oxygen or ozone to form esters, as proved by a peak at 289.3 eV in the C1s spectrum (Fig. 2e, bottom). The presence of hydroxyl groups might be attributed to the interaction of UV light with air, which not only generates ozone but also produces hydrogen radicals or peroxides. These reactive species can readily interact with oxygen atoms in MeO-TPD, facilitating the formation of OH groups[12]. These findings support the conclusion that UV irradiation induces photo-oxidative shrinking, a process characterized by bond dissociation and thickness dependency. The emergence of UV-induced O−C = O and (C = O)−O−(C = O) bonds suggests oxidative processes and potential dimerization[30]. Such oligomers further contribute to film shrinkage due to their increased molecular weight. Leveraging the intact benzene rings, the distant placement of reactive groups from the conjugated core, and the UV-resilient p-dopant, the conductivity of the HTL can be well preserved, concurrent with the UV-induced film thinning.

Raman spectra are obtained for MeO-TPD films with an initial thickness of 150 nm following 0.5 h and 1.5 h of UV irradiation. Normalization cannot be performed due to varying film thickness. As shown in Fig. 2f, Raman bands at 916 cm$^{-1}$, 1320 cm$^{-1}$, and 1568 cm$^{-1}$

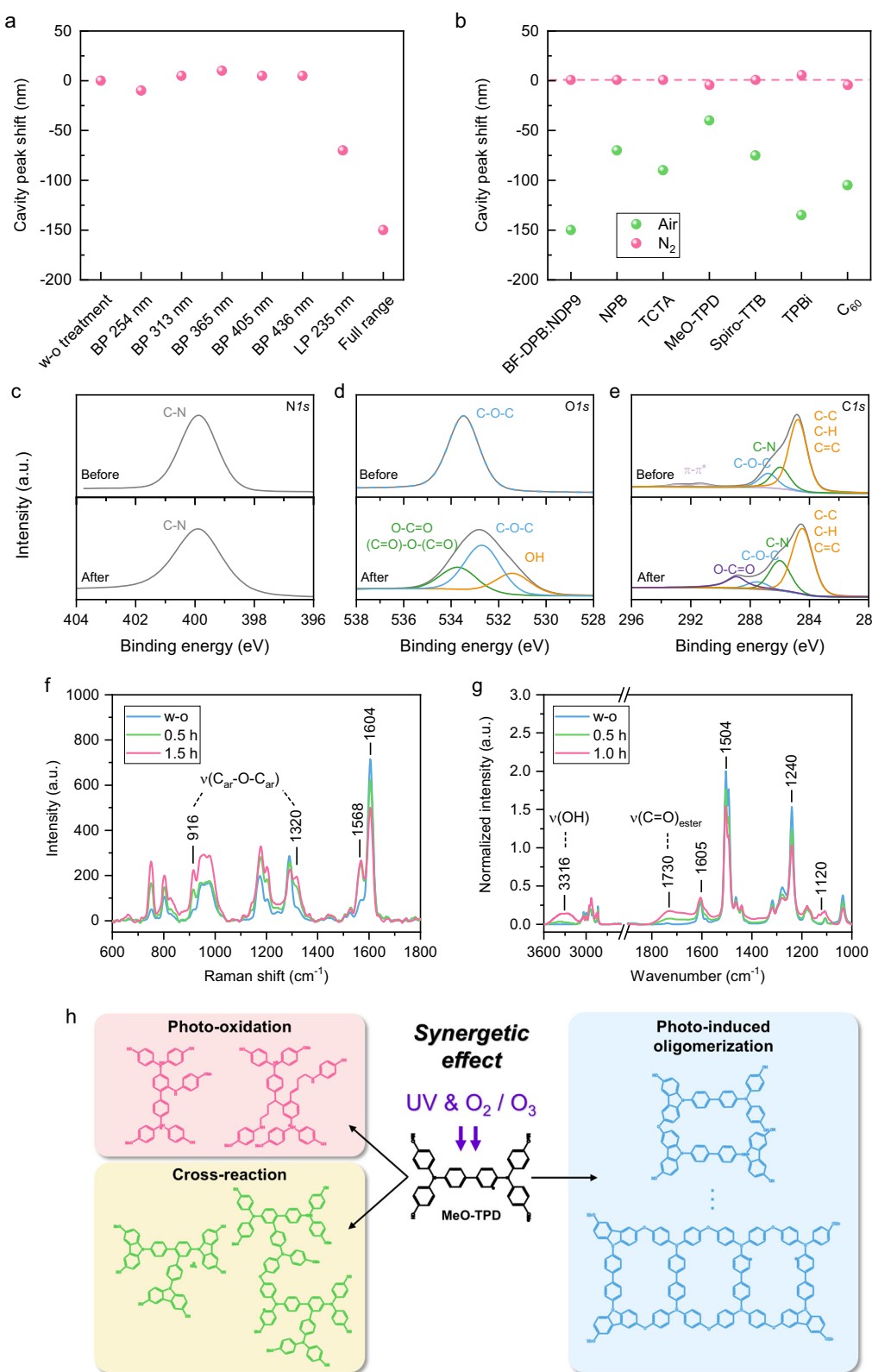

increase significantly after irradiation, corresponding to the formation of C(ar)−O−C(ar) bonds (potential dimerization of MeO-TPD molecules), and potential increase of the quinoid form of the molecule (1568 cm⁻¹: ring vibration of quinoid)[31]. The preservation of the most intense Raman band at 1604 cm⁻¹ (C−C ring and C−C−H bending vibrations of C(ar)) after 0.5 h UV irradiation confirms the integrity of aromatic rings, which is crucial for maintaining efficient charge transport pathways. Attenuated total reflection FTIR (ATR-FTIR)

spectroscopy is conducted on UV-treated MeO-TPD films with an initial thickness of 50 nm (Fig. 2g). The spectra are normalized to film thicknesses of 49 nm (pristine), 33 nm (0.5 h of irradiation), and 11 nm (1 h of irradiation), enabling a semi-quantitative analysis of band intensities. Upon UV irradiation, increase of aliphatic C−O−C groups (1120 cm⁻¹) supports the formation of dimers, while bands at 1730 cm⁻¹ and 3316 cm⁻¹ are particularly indicative of ester bond formation and forming of OH-groups supporting oxidative processes[32,33]. Meanwhile,

**Fig. 2 | Prerequisites and mechanism of the UV-induced thickness thinning effect. a** Absorption peak shift of BF-DPB:NDP9 (10 wt.%, 150 nm) layer exposed to various UV wavelengths for 2 h in air within a Cr (3 nm) / Au (60 nm) / BF-DPB:NDP9 (10 wt.%, 150 nm) / Ag (25 nm) cavity structure. The absorption peak of samples without treatment (w/o) is also presented. BP and LP denote bandpass and longpass filters, respectively. **b** Absorption peak shift of different organic materials including BF-DPB:NDP9 (10 wt.%), alpha-NPD, TCTA, MeO-TPD, Spiro-TTB, TPBi, and $C_{60}$ after 2 h UV irradiation in $N_2$ and air atmosphere, compared to samples without treatment (dashed line). All samples are within a Cr (3 nm) / Au (60 nm) / organic layer (150 nm) / Ag (25 nm) cavity structure. **c** The $N1s$, (**d**) $O1s$, and (**e**) $C1s$ peaks in XPS spectra of MeO-TPD (40 nm) on ITO substrates before and after 0.5 h UV irradiation. **f** Raman spectra of MeO-TPD (150 nm) w/o and after 0.5 h and 1.5 h of UV irradiation. **g** ATR-FTIR spectra of MeO-TPD w/o (49 nm film thickness) and after 0.5 h (33 nm film thickness) and 1.0 (11 nm film thickness) of UV irradiation. FTIR spectra are normalized to the layer thickness. **h** Schematic illustration of UV-induced chemical reaction in MeO-TPD. Left: oxidative fragments and products by photo-oxidation and cross-reaction. Right: ether-linked dimers and tetramers by photo-induced oligomerization.

the preserved stretching vibration bands at 1240 $cm^{-1}$ (alkyl-aryl ether groups) and 1504 $cm^{-1}$/1605 $cm^{-1}$ (aromatic rings) demonstrate again the retention of molecular structural integrity after 0.5 h of UV irradiation. To further support the proposed synergetic mechanism, MALDI-TOF is performed on MeO-TPD powders subjected to UV exposure (Supplementary Fig. 16). As schematically illustrated in Fig. 2h, the major products include oxidized fragments (m/z 340–500), cross-reacted larger molecules (m/z 850–1420), and ether-linked dimers and oligomers (~m/z 1074–1081). These chemical structures align with the representative MALDI-TOF peaks observed after UV treatment, supporting the simultaneous activation of photo-oxidation and oligomerization pathways, in line with prior spectroscopic observations. For BF-DPB thin films, ATR-FTIR results reveal similar UV-induced effects with absorption band positions comparable to the bands of MeO-TPD after UV treatment (Supplementary Fig. 17). These findings suggest that photo-oxidative processes are independent of the presence of oxygen atoms in HTL's molecular structure. Additionally, the shared chemical structures—such as conjugated aromatic backbones and electron-rich moieties, and the consistent thickness-thinning effect observed across investigated HTLs support the assumption that the UV-induced mechanism is generally applicable. Through surface-sensitive XPS analysis and bulk investigations using Raman and FTIR spectroscopy, we demonstrate that oxidative effects extend beyond the surface into deeper layers. However, the intensity of these effects diminishes with depth, consistent with the strong UV absorption (Supplementary Fig. 18) and scattering. Moreover, UV-C light generates ozone and decomposes it into highly reactive but short-lived species (O, •OH)[34,35], which further confines reactions to the surface. These combined factors enable precise control over thickness and optical properties without significantly altering bulk transport, which is crucial for maintaining the performance and stability of optoelectronic devices.

To evaluate the influence of UV-induced photoproducts on electronic properties, additional photoluminescence (PL) and time-correlated single-photon counting (TCSPC) measurements are performed on BF-DPB:NDP9 (10 wt.%, 100 nm) before and after UV treatment (Supplementary Fig. 19). The PL spectra show only a slight reduction in intensity, in line with minor absorption changes. Interestingly, a modest increase in fluorescence lifetime, from 6.86 ns (w-o treatment) to 7.31 ns (0.5 h of UV irradiation), is revealed by TCSPC, indicating that no additional non-radiative recombination channels are generated by the UV treatment. This lifetime change may result from the removal of quenching molecular fragments during UV exposure. The previously observed uniform surface roughness further confirms that no photoproduct aggregation or crystallization occurs that would hinder charge transport. Alongside XPS, Raman, and FTIR findings—showing preserved aromatic backbones and modification of peripheral groups—and stable conductivity, these results support the conclusion that UV-induced photoproducts do not impair charge transport in the treated films.

## UV-induced cavity tuning in OPDs
To illustrate the adaptability of this mechanism to organic electronics, NIR OPDs using a microcavity architecture are first fabricated. A scheme of the device architecture is shown in Supplementary Fig. 20a. Details on the layer sequence are provided in Supplementary Table 2. As the resonance wavelength is proportional to the cavity thickness, by varying the HTL thickness through UV irradiation, the desired detection wavelength can be tuned. To comprehensively investigate the UV approach in the device, films of BF-DPB:NDP9 (10 wt.%) with layer thicknesses of 110 nm and 445 nm are employed to achieve the first- and second-order resonance with electron transport layer (ETL) thicknesses of 90 nm and 105 nm, respectively. $C_{60}$: 2,2',6,6'-tetra-thienyl-4,4'-bithiopyranylidene (D6) (5 wt.%, 75 nm) serves as the photoactive layer owing to their broad charge-transfer state absorption band in the NIR range (Supplementary Fig. 21)[36], facilitating the exploration of thickness reduction effects on device performance. For the device fabrication, after evaporating the BF-DPB:NDP9 layer (10 wt.%, 110 nm or 445 nm), the samples are exposed to UV irradiation for variable durations in air. The process is continued with the deposition of the remaining layers under vacuum to finalize the devices (Fig. 1b).

The morphology and the energy level of the HTL material are first characterized prior to device fabrication. A smooth layer with an RMS roughness consistently below 0.6 nm is observed both before and after 1.5 h of UV treatment (Supplementary Fig. 8). The highest occupied molecular orbital (HOMO) level is also well preserved, as proved by ultraviolet photoelectron spectroscopy (UPS) (Supplementary Fig. 22). The external quantum efficiency ($EQE_{PV}$) of UV-irradiated OPD is presented in Fig. 3a, b. For untreated OPD, the first-order resonance peak appears at 1185 nm with a narrow full-width-at-half-maximum (FWHM) of 34 nm. Balancing between the initial thickness and the thinning rate of BF-DPB:NDP9 (10 wt.%) under UV exposure, a treatment time of 4 h is identified to ensure the retention of the HTL for a well-performing device. Consequently, a resonance wavelength of 945 nm is obtained, with a well-preserved cavity feature (FWHM ≤ 30 nm). The blue shift in the resonance peak visually proves the reduction in HTL thickness under UV irradiation. Due to a higher absorption coefficient of $C_{60}$:D6 blend at shorter wavelengths, the $EQE_{PV}$ slightly improves. Comparable findings are likewise observed in the second-order cavity devices[36,37]. The expected $EQE_{PV}$ profile and peak values verify the HTL's efficacy in facilitating hole transport subsequent to UV treatment. $EQE_{PV}$ is further enhanced to 10.4% under −3 V after 2 h of UV treatment (Supplementary Fig. 23), and reaches up to 72% under UV exposure in broadband DCV2-5T-Me(3,3):$C_{60}$-based OPDs at 0 V (Supplementary Fig. 24), demonstrating compatibility with high-performance systems. Considering the thickness reduction of the HTL, the dark current density ($J_d$) of the UV-irradiated OPD is increased under reverse bias within a reasonable range (Supplementary Fig. 25). Notably, in the case of OPDs possessing a 445 nm thick HTL, the impact on $J_d$ is mitigated, which is ascribed to a mild change in the HTL thickness, which consequentially leads to a negligible variation in the shunt resistance ($R_{sh}$) following the UV exposure.

To evaluate the lowest detectable limit of our UV-irradiated OPDs, noise measurements are conducted at 0 V (Supplementary Fig. 26), from which the specific detectivity ($D^*$) is derived. As depicted in Fig. 3c, d, the real specific detectivity ($D_{real}^*$) is comparable to the theoretical thermal noise-limited detectivity

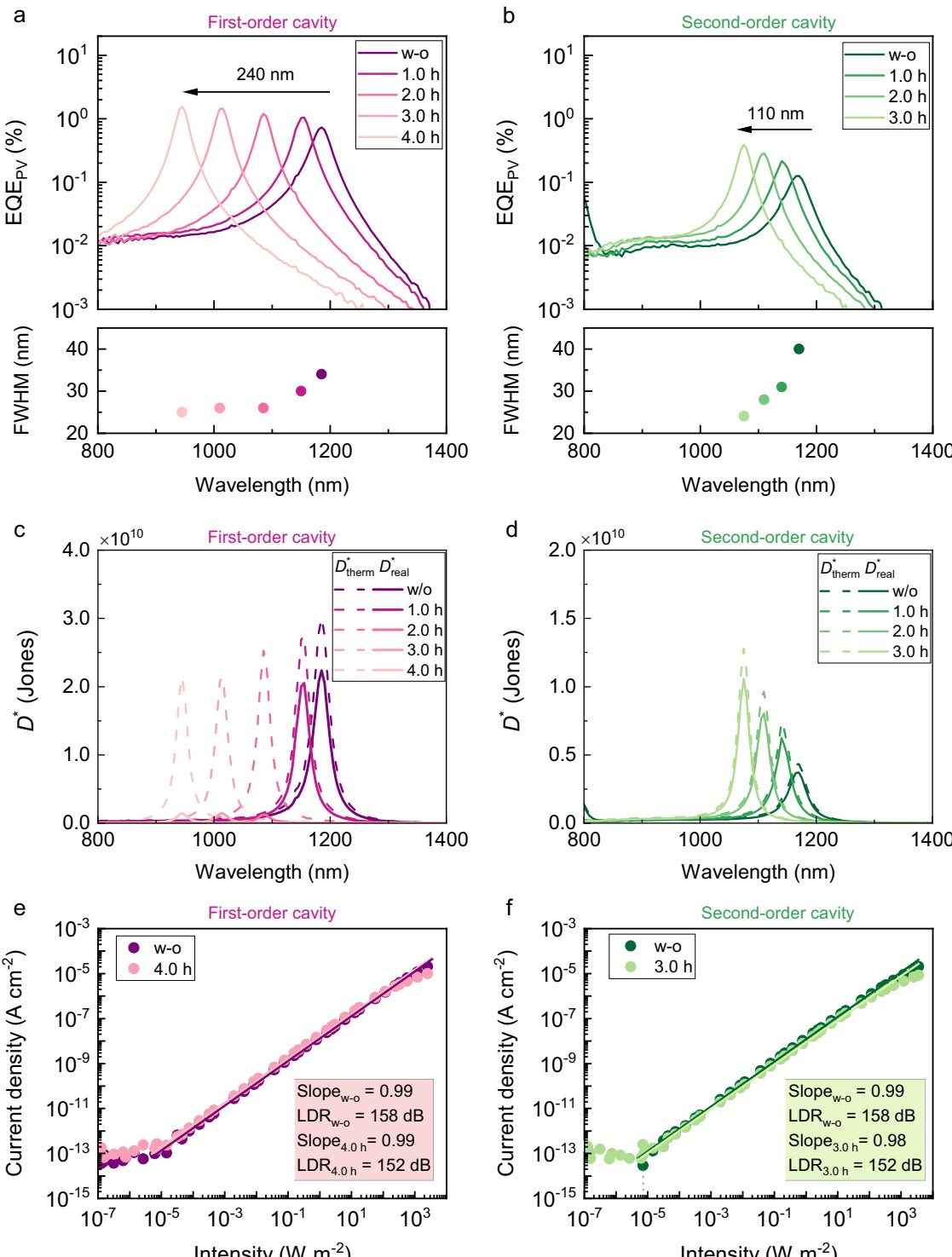

**Fig. 3 | Characterizations of the UV-treated OPDs.** $EQE_{PV}$ spectra and corresponding FWHMs of UV-treated OPDs at the (**a**) first and (**b**) second-order cavities. Specific detectivity spectra of UV-treated OPDs at the (**c**) first and (**d**) second-order cavities. Dashed lines and solid lines represent the calculated thermal noise-limited $D^*$ ($D^*_{therm}$) and real $D^*$ ($D^*_{real}$) at short-circuit conditions, respectively. Linear dynamic range of UV-treated OPDs at the (**e**) first and (**f**) second-order cavities under 455 nm light irradiation at 0 V. The BF-DPB:NDP9 (10 wt.%) layer is 110 nm or 445 nm in the respective OPDs prior to UV treatment.

($D^*_{therm}$) at short-circuit conditions (assuming thermal noise only) except for the devices with thinner HTLs, confirming thermal noise as the dominant noise contribution. Peak $D^*_{real}$ on the order of $10^9$ to $10^{10}$ Jones are achieved in the spectral range of 900–1200 nm for both UV-irradiated OPDs. For first-order cavity devices with reduced HTL thickness, $J_d$ steadily increases due to

the enhanced charge carrier injection, leading to a lower $D^*_{real}$. Conversely, when the HTL is optimally thick, the UV treatment can couple a tunable spectrum with device stability without performance loss, as demonstrated by OPDs built on second-order cavities. The performance metrics are compiled in Supplementary Table 3. Owing to the intrinsic properties of the

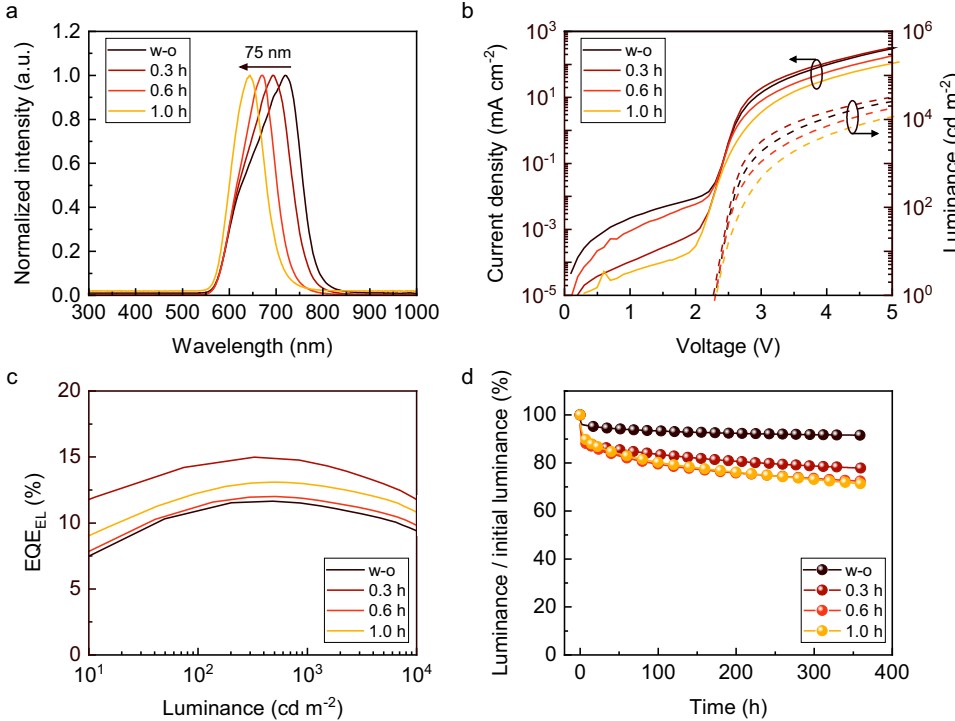

**Fig. 4 | Characterizations of the UV-treated OLEDs. a** Normalized electroluminescence (EL) spectra of OLEDs with varied UV irradiation time. The spectra are measured at a viewing angle of 0°. **b** Current density-voltage characteristics and irradiance values of UV-treated OLEDs. **c** $EQE_{EL}$ of UV-treated OLEDs measured in an integrating sphere mode. **d** Lifetime performance of UV-treated OLEDs at a fixed driving current of 1 mA.

material system employed, the $D^*$ of the UV-irradiated OPDs is consistent with previous studies[36]. Notably, the reported $D^*_{real}$, based on real noise measurements, provides a more reliable basis for fair performance comparisons[38,39]. Moreover, the $D^*_{real}$ of broadband DCV2-5T-Me(3,3):C$_{60}$-based OPDs is significantly enhanced to above $10^{13}$ Jones at 0 V after 0.5 h of UV treatment due to reduced $J_d$ (Supplementary Fig. 24). The detectivity ranks one of the highest for OPDs, demonstrating the strong compatibility and effectiveness of the UV treatment. An OPD with a top-illuminated n-i-p architecture is also fabricated, in which the HTL is on top for direct UV exposure. The approach remains mechanistically effective (Supplementary Fig. 27), as indicated by decreased dark current. However, due to degradation of the underlying layers, the $EQE_{PV}$ is reduced. This contrast highlights the stability and practical advantage of applying the UV treatment to p-i-n architectures, which are highly relevant to industrial applications.

The linear dynamic range (LDR) at zero bias is assessed and illustrated in Fig. 3e, f. Due to the insufficient intensity of the NIR LED to cover the expected intensity range, the 455 nm LED, corresponding to the second-order resonance peak of each OPD, is selected to ensure an accurate LDR measurement. All the narrowband OPDs within 900 – 1200 nm exhibited nearly eight orders of magnitude of linear response under light irradiation, denoted by LDR > 152 dB. The large linearity underlines the effective charge transport even after UV irradiation. Furthermore, 4 h of UV irradiation results in a slightly lower sub-linearity onset (> 576 W m$^{-2}$) of OPDs compared to untreated ones, indicating that the UV treatment has only a minimal impact on the recombination rate at high light intensities. We further explored the response speed of our OPDs by conducting transient photocurrent measurements (Supplementary Fig. 28). The −3 dB cut-off frequencies ($f_{-3dB}$) of 368 KHz and 777 KHz for first and second-order cavity OPDs are also demonstrated under 455 nm light irradiation, after being subjected to 4 h of UV exposure. These values significantly exceed the speed requirements of advanced imaging applications[40].

## UV-induced cavity tuning in OLEDs

Building on the successful demonstration of efficient cavity tuning in UV-irradiated OPDs, we extended this UV treatment strategy to precisely tune the emission spectrum of OLEDs. A series of top-emitting red OLEDs with a cavity architecture—an approach commonly used in smartphones—is carefully designed and fabricated using a fully vacuum vapor deposition process. The device architecture is Cr (3 nm) / Au (80 nm) / Spiro-TTB:NDP9 (7 wt.%, 55 nm) / NPB (10 nm) / NPB: iridium(III)bis(2-methyldibenzo-[f,h]chinoxalin)(acetylacetonat) (Ir(MDQ)$_2$(acac)) (10 wt.%, 20 nm) / 4,7-diphenyl-1,10-phenanthroline (BPhen) (10 nm) / BPhen:Cs (1:1, 65 nm) / Au (2 nm) / Ag (19 nm) / NPB (82 nm), see Supplementary Fig. 20b. Ir(MDQ)$_2$(acac) is used as the phosphorescent emitter to harvest triplet excitons. Au (2 nm) / Ag (19 nm) is employed as the transparent top electrode, and the 82 nm NPB capping layer is introduced to improve light outcoupling. The HTL, composed of Spiro-TTB:NDP9, undergoes UV irradiation from 0.3 to 1.0 h. Following this treatment, the remaining layers are deposited under vacuum to complete the device fabrication (Fig. 1b).

As indicated by AFM analysis (Supplementary Fig. 29), an RMS roughness of less than 0.8 nm throughout 1.5 h of UV treatment ensures a smooth layer for device fabrication. We notice that the emission peak of the red OLEDs gradually blue-shifts by 75 nm—from 718 nm to 693 nm, 668 nm, and finally 643 nm—after 0.3, 0.6, and 1 h of UV irradiation, respectively (Fig. 4a). This shift corresponds to an approximate 22 nm reduction in the HTL thickness (Supplementary Fig. 30), resulting in a shorter resonance wavelength. Also, the concurrent decrease in angular shift with increasing UV exposure further illustrates the progressive thinning of the HTL (Supplementary Fig. 31). As the HTL gets thinner, the microcavity effect within the OLED structure is reduced, resulting in a more uniform emission

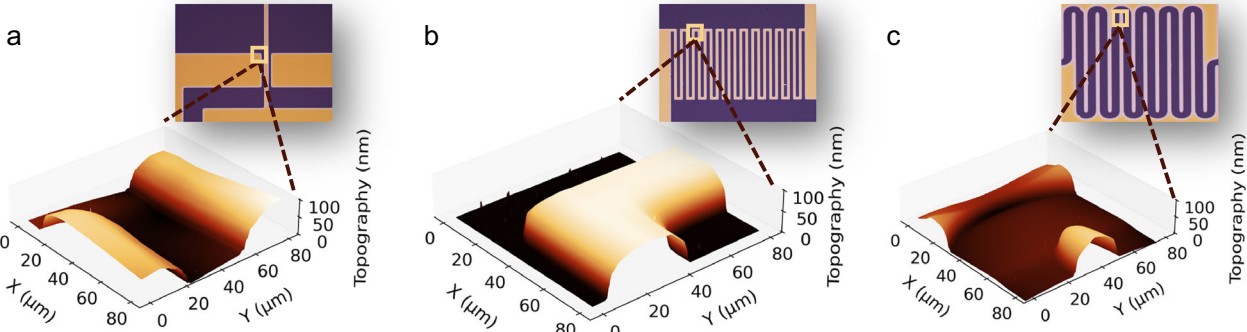

**Fig. 5 | Micrometer-scale organic patterns induced by UV irradiation.** Microscope images and AFM scans of organic patterns after UV treatment with (**a**), (**b**) linear edges and (**c**) curved edges. In the microscope images, the remaining BF-DPB appears yellow, and the navy regions correspond to Si. The scan area for the AFM measurement is indicated by the yellow square. The AFM color scale is linearly distributed between the minimal and maximal heights indicated. The UV-induced structure pattern closely mirrors the designed resolutions of (**a**) 40 μm, (**b**) 40 μm and (**c**) 25 μm.

spectrum across different viewing angles. The current density-voltage-luminance characteristics of the red OLEDs are illustrated in Fig. 4b. With 0.3 h of UV treatment, both the current density and luminance of OLEDs at a certain voltage are enhanced compared to the untreated device. As the UV treatment duration is extended to 0.6 and 1.0 h, the devices continue to exhibit robust performance with luminance levels remaining above $10^4$ cd m$^{-2}$. Although there is a slight decrease in current density and luminance at a certain voltage, this subtle change underscores the stability of the device's emission characteristics even under prolonged UV exposure. The $EQE_{EL}$ of the UV-irradiated OLEDs, measured in an integrating sphere, is plotted in Fig. 4c. The $EQE_{EL}$ of the devices before and after UV treatment remains relatively stable, ranging from 12% to 15%. The highest $EQE_{EL}$ is given by the red OLED subjected to 0.3 h UV irradiation. Since the energy levels remain unchanged (Supplementary Fig. 22), the improvement is likely due to enhanced charge injection and increased photon flux within the optical cavity, resulting from the reduced HTL thickness. This interpretation is further supported by devices with physically varied HTL thicknesses showing a similar $EQE_{EL}$ trend (Supplementary Fig. 32).

To estimate the impact of UV treatment on devices stability, we measure the lifetime of OLEDs under different treatment conditions. As presented in Fig. 4d, the untreated OLED shows the highest stability, retaining over 90% of its initial luminance after 350 h. For the UV-irradiated red OLEDs, a reasonable decline in lifetime with increasing UV exposure duration is observed. Specifically, the $LT_{80}$ values for UV-treated OLEDs are obtained at 231 h, 105 h and 91 h following 0.3 h, 0.6 h and 1.0 h of UV exposure, respectively. Interestingly, the lifetime performance of the devices treated for 0.6 and 1.0 h is quite similar, with the difference between these two being much smaller than the change observed between the untreated device and the one treated for 0.3 h. This suggests that while moderate UV exposure (up to 0.3 h) induces notable changes in the device stability, further increases in UV treatment time have a diminishing impact on lifetime. This effect implies that the primary structural adjustments within the HTL occur within the first 0.3 h of UV exposure, after which the material stability properties reach a relatively consistent state. The phenomenon provides a precise and broad optimization window, enabling performance improvements and a large spectrum tuning without substantially compromising device stability. Moreover, in an optimized OLED system with a long lifetime, such as exceeding 30000 h[41,42], the slight reduction in lifetime observed with UV treatment might be negligible, further enhancing the practical value of this approach. The robust performance of UV-treated OPDs and OLEDs clearly demonstrates that the surface morphology, hole transport capability, and work function of the HTLs remain unaffected, underscoring the reliability of the approach for high-performance optoelectronic devices.

## Micrometer-scale organic patterns induced by UV irradiation

Finally, we investigate the structural effects of UV treatment at the micrometer scale—a level of precision that, while achievable with fine shadow masks in the 20–50 μm range, provides enhanced flexibility and precision for pixelated integration of organic optoelectronic devices. To demonstrate this, we evaporated multiple BF-DPB films (100 nm) onto Si substrates, followed by UV exposure through three different photolithography masks. The use of photolithography masks is primarily due to the broad exposure area of the UV lamp, which limits direct patterning resolution. However, our method is readily adaptable to a maskless configuration if a UV laser beam emitting at 185 nm is employed and sufficiently focused, enabling even finer lateral control and greater patterning flexibility. The resulting patterns are characterized using optical microscopy and AFM, and presented in Fig. 5. The UV treatment produces a structure pattern closely matching the designed resolution, with both linear and curved-edge patterns exhibiting sharp contrast and resolutions as fine as 25 μm. Supplementary Fig. 33 provides further details on these structured patterns. To further highlight the resolution capability of the method, patterns as small as 8.3 μm (mask of 8 μm) and 6.2 μm (mask of 5 μm) are achieved using photolithography masks and shadow masks, respectively (Supplementary Fig. 34), suggesting the method's suitability for even sub-10 μm features.

Given that pixel dimensions in advanced OLED displays typically range from 30 to 100 μm, the ability to consistently produce sub-30 μm features with UV treatment is a significant advancement. This fine patterning capability demonstrates the potential for next-generation, high-resolution organic electronics, offering a scalable, mask-free method for precise pixel control at finer scales than traditional shadow masks can achieve, particularly in compact devices. Moreover, the inherent versatility of UV treatment allows for rapid prototyping and seamless adaptation to different patterning requirements without the need for costly, custom-fabricated metal masks, making it ideal for research and commercial production alike. This flexibility, coupled with its high resolution, positions UV treatment as a promising solution for the evolving demands of organic optoelectronic device manufacturing, where precise pixel-level integration is critical for optimizing both performance and miniaturization.

We report a method for treating organic small molecule HTLs using UV irradiation in an atmosphere. As exposure to UV irradiation intensifies, a uniform thickness-thinning phenomenon is activated, driven by the interplay of photoinduced oligomerization and photo-oxidative layer shrinking, as proven by XPS, MALDI-TOF, Raman and FTIR measurements. Notably, the conductivity of the HTL is not sacrificed irrespective of doping concentrations. This approach shows full-thickness tunability and is also applicable in a wide variety of organic materials. Based on this concept, UV-treated NIR OPDs and UV-treated red OLEDs in cavity architectures are fabricated, both show-casing remarkable tunability and performance metrics comparable to the traditional thickness tuning method. Furthermore, the potential of UV irradiation extends to the realm of structured organic HTLs, enabling finely controlled architectures with (sub)-nanometer precision. While our work opens new perspectives for device engineering and integration into state-of-the-art organic electronics, further studies on long-term stability and device lifetime are essential before practical implementation.

## Methods

### Device preparation
The layers of the OPDs and OLEDs are thermally evaporated at ultra-high vacuum (base pressure of $<10^{-7}$ mbar) on pre-cleaned glass substrates. After the evaporation of the BF-DPB:NDP9 layer (OPD) and Spiro-TTB:NDP9 (OLED), the devices are irradiated by UV for variable durations in air. The following layers are then deposited to complete the fabrication of the devices. Details on the layer sequence for each device are listed in Supplementary Table 2. All organic materials are purified 1–2 times via sublimation. The device area is defined by the geometrical overlap of the bottom and the top contact and is equal to 6.44 mm$^2$. To avoid exposure to ambient conditions, the organic part of the device is covered by a small glass substrate, which is glued on top.

### Spectrum measurements
The spectrum of the UV lamp is measured by a miniature spectrometer (RGB Photonics Qmini, Germany), which covers a wide range from 190 to 1100 nm in a single device.

### Conductivity measurements
The organic film is deposited onto a glass substrate, followed by the evaporation of Au interdigitated electrodes (111 × 0.5 mm). Conductivity is determined by measuring the current between adjacent electrode contacts:

$$\sigma = \frac{bI}{Uhl} \tag{1}$$

where $b$ is the distance between two contacts (0.5 mm), $l$ the length of the contacts (111 mm), $h$ the film thickness, $U$ the external voltage, and $I$ the measured current.

### Ellipsometry measurements
Variable-angle spectroscopic ellipsometry is performed with an M2000 UI (J.A. Woollam Co., Inc., Lincoln, USA, wavelength range: 245–1690 nm). An optical model is fitted for the layer stack Si / SiO$_2$ / organic film / roughness layer / ambient. For Si and SiO$_2$ optical dispersions are taken from the database (CompleteEASE, J.A. Woollam Co., Inc.), the ambient is air. The organic film optical constants are fitted to oscillator models with Tauc-Lorentz and Gaussian oscillators. The roughness layer is fitted by an effective medium approximation with 50% void. BF-DPB and BF-DPB:NDP9 layers have uniaxial optical anisotropy, while energy positions and band width in the uniaxial oscillator model are coupled (z to xy).

### Absorption measurements
The transmission ($T$) and reflection ($R$) spectra in the integral sphere mode are acquired with a laboratory UV-VIS-NIR spectrometer (Shimadzu SolidSpec-3700, Japan). The depicted absorption is calculated from:

$$A = 1 - R - T \tag{2}$$

### Current-voltage characteristics
For OPDs, the current-voltage characteristics in the light are performed at an intensity of 1000 W m$^{-2}$ utilizing a xenon lamp (Ushio UXL-300D-0, Japan) in a sun simulator (Solar Light Co. Sunlight simulator 16S-003-300-AM1.5, USA). The intensity is calibrated to a Si photodiode (Hamamatsu Photonics S1337, Japan). The dark $J$-$V$ curves are further characterized with a high-resolution SMU (Keithley Instruments Keithley 2635 A, USA). Every measurement data point is acquired after the steady-state condition is achieved.

### $EQE_{PV}$ measurements
The $EQE_{PV}$ spectra are measured with a lock-in amplifier (Signal Recovery SR 7265, USA) under monochromatic illumination (Oriel Xe Arc-Lamp Apex Illuminator combined with Newport Cornerstone 260 1/4 m monochromator, USA) using a calibrated monocrystalline Si reference diode (Hamamatsu S1337, Japan, calibrated by Fraunhofer ISE). A mask is used to minimize edge effects and define an exact photoactive area (2.997 mm$^2$).

### Sensitive $EQE_{PV}$ measurements
The sensitive $EQE_{PV}$ measurement is conducted in air using a 250 W halogen lamp (OSRAM HLX 64657, Germany), which is modulated at 170 Hz and directed through a double monochromator (Quantum Design MSHD-300A, Germany). The monochromatic light is focused onto the OPD, and the photocurrent generated by the OPD is amplified with a current-voltage preamplifier (Stanford Research Systems SR 570, USA) and then fed into a lock-in amplifier (Stanford Research Systems SR830, USA) at short-circuit conditions. The photon flux is determined using calibrated photodiodes: a Si photodiode (Thorlabs FDS100-CAL, USA) and an InGaAs photodiode (Hamamatsu Photonics G12183_020K, Japan). The $EQE_{PV}$ is calculated by dividing the photocurrent of the OPD by the measured photon flux.

### Linear dynamic range
The light intensity of a 455 nm LED (Thorlabs M660L4, USA) driven by an LED driver (Mightex Systems BLS-1000-2, Canada). To achieve a wide light intensity range, a series of neutral density filters (Thorlabs, USA) is utilized. The sample is measured in an electrically shielded box to decrease external noise sources. The photocurrent is then pre-amplified (FEMTO Messtechnik DLPCA-200, Germany) and provided to a lock-in amplifier (Stanford Research Systems SR865A, USA). The signal is calibrated to a Si diode (Thorlabs SM05PD3A, USA). The LDR is calculated using the equation:

$$LDR = 20 \log\left(\frac{P_{max}}{P_{min}}\right) \tag{3}$$

where $P_{max}$ and $P_{min}$ are the highest and lowest light power among which the measured photocurrent of the devices shows a linear relation with the incident light power.

Noise spectral density. The OPD is measured at 0 V in an electrically shielded box to provide darkness and disentangle the real noise data from the ambient environment and electrical artefacts. The noise current is amplified by a very low-noise current-voltage preamplifier (FEMTO Messtechnik LCA 30-1 T, Germany) and recorded by a high-speed, low-noise oscilloscope (Tektronix DPO7354C, USA). With the

$EQE$ and measured noise spectral density ($i_n$), the $D_{real}^*$ of the device can be calculated based on the equations:

$$D_{real}^* = \frac{e\lambda EQE\sqrt{A}}{hci_n} \quad (4)$$

where $e$ is the elementary charge, $\lambda$ the incident wavelength, $A$ the device area, $h$ the Planck constant and $c$ the speed of light. The theoretical $D_{therm}^*$ is derived from the white noise ($i_w$), which comprises the shot noise ($i_s$) and thermal noise ($i_t$), as defined by the following equations:

$$i_w = \sqrt{i_s{}^2 + i_t{}^2} = \sqrt{2eI_d\Delta f + \frac{4k_BT\Delta f}{R_{sh}}} \quad (5)$$

where $I_d$ is the dark current, $\Delta f$ the bandwidth, $k_B$ the Boltzmann constant, and $T$ the temperature.

### −3 dB cut-off frequency
The light intensity of a 455 nm LED (Thorlabs M660L4, USA) is modulated at frequencies up to 4 MHz. The photocurrent is then preamplified (FEMTO Messtechnik DHPCA-100, Germany) and provided to a lock-in amplifier (Stanford Research Systems SR865A, USA). The signal is calibrated to a Si diode (Thorlabs SM05PD3A, USA).

### OLED Characterizations
The EL spectra are measured using a spectrometer (Instrument Systems CAS140CT, Germany) and a calibrated Si photodiode. The current-voltage-luminance characteristics of the OLEDs are obtained using an integrating sphere (Labsphere Inc. LMS-100, USA), with a source measure unit (Keithley Instruments Keithley 2400, USA) driving the devices and recording their electrical performance. A pico-amperemeter (Keithley Instruments Keithley 6400, USA) captures the photodiode signal integrated within the sphere, enabling precise measurements of luminescence at different voltages. This setup allows for the accurate determination of absolute $EQE_{EL}$ values. The angular-resolved spectral radiant intensity is measured using a custom-built goniometric setup, with samples mounted in a rotating holder. A laser aligns the center of rotation of the OLED with the optical axis, and a spectrometer (Ocean Optics USB4000, Germany) records the spectral irradiance proportional to the radiant intensity. The OLED lifetime is recorded by an automated measurement system.

### TCSPC measurements
Fluorescence decay measurements are carried out using a custom-built TCSPC system, with a 375 nm diode laser (PicoQuant LDHDC375) as the excitation source, which is controlled by a laser driver (Pico-Quant PDL820). The signal is detected by a hybrid photomultiplier detector (PicoQuant PMA Hybrid 40), connected to a TCSPC module (PicoQuant TimeHarp 260). The emission is collected at a right angle to the excitation beam, with the emission wavelength selected by a monochromator Princeton Instruments SpectraPro HRS-300) and detected by a CCD (Princeton Instruments PIXIS 100B 1340×100 pixels).

### XPS and UPS measurements
40 nm layers of MeO-TPD, BF-DPB:NDP9 and Spiro-TTB:NDP9 are deposited onto a gold foil substrate under high vacuum conditions (-10⁻⁸ mbar). The measurement chamber is directly connected to the evaporation chamber, making an immediate measuring possible without exposing the sample to atmospheric conditions. The base pressure within the measurement chamber is maintained at $4 \times 10^{-10}$ mbar. The measurement is performed using a Phoibos 100 system equipped with an HSA3500 power supply. For XPS measurements, an XR-50 X-ray source (Al-anode, with an ionization energy of approximately 1500 eV, Specs, Germany) and for UPS measurements a Helium discharge lamp (UVS 10/35, Specs, Germany) is used. UPS and XPS measurements are performed on identical but separately prepared samples. The analyzer is calibrated with ultraviolet photoelectron spectroscopy to the Fermi edge of sputter-cleaned silver. Following the initial measurement, the samples are transferred in a nitrogen atmosphere to the UV treatment setup to minimize potential contamination. After 0.5 h of UV irradiation, the sample is returned to vacuum conditions for the post-UV analysis. To overcome the influence of sample charging during the measurements, the XPS spectra are subsequently calibrated to the C1s peak, assuming it at a constant position of 284.8 eV[43].

### AFM measurements
A Bruker Dimension Icon atomic force microscope is used to obtain the precise shape of the UV-generated patterns. The measurement is performed in tapping mode with ScanAsyst using RTESPA-150 tips. Background subtraction of the data is processed with Gwyddion.

### Film thickness measurements
The film thicknesses under ozone-only conditions and the microstructure pattern are evaluated by a profilometer (Veeco Instruments Dektak 150, USA). The instrument utilizes a 5 μm radius stylus, which is moved across the surface of the sample under a controlled load of 3 mg. The scan length is set to 500 μm with a resolution of 0.083 μm to capture surface topology with high precision.

### HPLC-MS measurements
HPLC-MS is measured using an Agilent 1260 liquid chromatograph coupled to an Agilent 6538 ESI-Q-TOF mass spectrometer.

### Raman measurements
The Raman Imaging System alpha300R (WITEC GmbH, Germany) with a 20× Zeiss objective is used with an excitation laser wavelength of 532 nm and a laser power of 1 mW. The integration time is 0.5 s, and the number of accumulations is 500. All spectra are baseline corrected and smoothed using the Savitzky-Golay algorithm. The spectral resolution is 2 cm⁻¹. For each film, three Raman spectra at three different positions are averaged within the wavenumber range between 600 and 3665 cm⁻¹. The resulting Raman intensity is not normalized due to variable thickness and the ratio of chemical groups within films.

### FTIR measurements
ATR-FTIR spectroscopy using the advanced multiple ATR approach is performed on HTL deposited films on FZ silicon wafers with an initial thickness of 50 nm and after UV treatment at 0.5 h, 1 h, and 1.5 h. The FZ wafers also function as the ATR crystal with ca. 46 reflections. 500 scans per spectrum are acquired using an evacuated FTIR spectrometer Vertex 80 v (Bruker Optik GmbH, Germany) equipped with both an ATR Si wafer 40 mm unit (Bruker Optik GmbH, Germany) and a highly sensitive mercury cadmium telluride (MCT) detector (InfraRed Associates, Inc., USA) in the wavenumber range of 4000 – 1000 cm⁻¹ with a resolution of 4 cm⁻¹. Spectra are referenced to the spectrum of an empty FZ wafer and baseline corrected. As an additional post-processing, spectra are also normalized to the film thickness obtained by spectroscopic ellipsometry. Band assignments are performed using references in the main text[44].

### MALDI-TOF measurements
The mass spectrometry analysis is performed on a Bruker Autoflex Speed MALDI-TOF MS (Bruker Daltonics GmbH & Co. KG, Germany) using trans-2-[3-(4-tert-Butylphenyl)−2-methyl-2-propenylidene]malononitrile (DCTB) as matrix. Prior to testing, MeO-TPD is dissolved in Acetone.

## Data availability

The data supporting the findings of this study are available in this article (including the source data of each Figure) and its Supplementary Information files. All data are available from the corresponding author upon request. Source data are provided with this paper.

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

## Acknowledgments

This work is funded by the projects "FLEXMONIRS" (01DR20008A) and "Pergamon" (16ME0012) from the German Federal Ministry of Education and Research (BMBF). L.C.W. acknowledges the graduate academy project 2767 (GRK 2767) funded by the German Research Foundation (DFG). The authors acknowledge helpful discussions on chemical analysis with Martin Geisler (IPF).

## Author contributions

S.X., H.K., J.B. and K.L. proposed and supervised the project. S.X. and J.B. designed the experiments. S.X., L.C.W., J.W. and T.Z. performed the OPD characterization. S.X., V.P., Y.L., D.S., J.V., R.H. and S.R. conducted the OLED characterization. S.B. carried out the XPS measurements. A.H. and S.D.S. contributed to the AFM measurements. E.B., D.F. and M.M. analyzed photochemical processes from Raman and FTIR results. M.F.X.D. and F.O. simulated the theoretical absorption. M.W., G.L. and X.F. contributed to chemical structure analysis and helpful discussions. S.X., J.B., H.K. and K.L. interpreted the data and jointly wrote the manuscript. All authors commented on the manuscript.

## Funding

## Competing interests

The authors declare no competing interests.
