## [Transparent Peer Review file · Nature Communications]

Light-induced fine-tuning of optical cavities for organic optoelectronic devices

Corresponding Author: Professor Karl Leo

Version 0:

Reviewer comments:

Reviewer #1

(Remarks to the Author)

I believe the revised version of the manuscript addresses my criticisms in good level of detail, and I recommend the manuscript for publication in Nature Communications.

Reviewer #3

(Remarks to the Author)

The authors have responded to all my comments and questions. I recommend that the manuscript is accepted for publication.

Reviewer #4

(Remarks to the Author)

I note the careful reports from referees 1 - 3 and the thorough responses the authors have produced, and I think this process has been addressed these referees' concerns.

Two points:

This paper does present this UV treatment as a practical method for tuning OLED structures, and the implication is that it can be adopted in commercial industrial production. I think this is a big claim, and I share referee 2's concern about the effect of UV treatment on long-term device operational lifetime. I agree it is surprising that the UV treated layers are as good as they are, but they are not as good as untreated layers, so a lot of work to do before practically useful, and it would require a compelling performance advantage for industry to go to this trouble. I think the claims on usefulness may need to be toned down a bit.. For example, the statement in the conclusion: "the realization of meticulously crafted architectures with industry-compatible precision" implies that the technology is fully industry compatible.

I note that the etching rate is independent of film thickness (no reason to expect otherwise), so this is not a process that makes the hole transport more uniform in thickness, and this does limit the potential usefulness of this process...

Summary - this is interesting work, and it surprising that UV/oxygen etching is as good as is shown here. On balance I'd be happy to see it published - but I think the claims for usefulness need to be tempered, and more emphasis on the impact of degradation in performance.

Reviewer #1

I believe the revised version of the manuscript addresses my criticisms in good level of detail, and I recommend the manuscript for publication in Nature Communications.

Response: We thank for the reviewer's positive assessment and recommendation for publication.

Reviewer #3

The authors have responded to all my comments and questions. I recommend that the manuscript is accepted for publication.

Response: We are grateful for the reviewer's encouraging comments on our manuscript for publication.

Reviewer #4

I note the careful reports from referees 1-3 and the thorough responses the authors have produced, and I think this process has been addressed these referees' concerns.

Two points:

This paper does present this UV treatment as a practical method for tuning OLED structures, and the implication is that it can be adopted in commercial industrial production. I think this is a big claim, and I share referee 2's concern about the effect of UV treatment on long-term device operational lifetime. I agree it is surprising that the UV treated layers are as good as they are, but they are not as good as untreated layers, so a lot of work to do before practically useful, and it would require a compelling performance advantage for industry to go to this trouble. I think the claims on usefulness may need to be toned down a bit. For example, the statement in the conclusion: "the realization of meticulously crafted architectures with industry-compatible precision" implies that the technology is fully industry compatible.

Response: We thank the reviewer for the thoughtful and encouraging assessment of our work, as well as for the constructive suggestions on moderating our industrial claims and clarifying the practical limitations. We agree that while our

method shows promise, further optimization will be required before it can be considered for full industrial adoption. Accordingly, we have revised the conclusion to temper our claims. The relevant sentences on **page 16** now read:

Changes: “Furthermore, the potential of UV irradiation extends to the realm of structured organic HTLs, enabling finely controlled architectures with (sub)-nanometer precision. While our work opens new perspectives for device engineering and integration into state-of-the-art organic electronics, further studies on long-term stability and device lifetime are essential before practical implementation.”

I note that the etching rate is independent of film thickness (no reason to expect otherwise), so this is not a process that makes the hole transport more uniform in thickness, and this does limit the potential usefulness of this process.

Response: We thank the reviewer for this valuable comment. We agree that the UV thinning process does not intrinsically correct global thickness non-uniformities of a film. Rather, its strength lies in enabling precise and local lateral post-fabrication correction of organic layer thicknesses using techniques such as masks, scanning UV lasers, or maskless photolithography, thereby facilitating the realization of final uniformity by only treating the thick areas. To clarify this point, we have added the following sentence on **page 6** to the main text.

Changes: “In addition, the UV irradiation method enables selective treatment of thick regions, eliminating local inhomogeneities and improving overall film uniformity.”

Summary—this is interesting work, and it surprising that UV/oxygen etching is as good as is shown here. On balance I'd be happy to see it published - but I think the claims for usefulness need to be tempered, and more emphasis on the impact of degradation in performance.

Response: We greatly appreciate the reviewer's comment, which helps us clarify the presentation of our results.

We once again thank all reviewers for their time and constructive feedback on our work. With the revisions and clarifications provided, we believe that this manuscript is now suitable for publication in *Nature Communications*.